# Volatile Compound Abundance Correlations Provide a New Insight into Odor Balances in Sauce-Aroma Baijiu

**DOI:** 10.3390/foods11233916

**Published:** 2022-12-05

**Authors:** Qijie Guan, Lian-Jun Meng, Zilun Mei, Qingru Liu, Li-Juan Chai, Xiao-Zhong Zhong, Lei Zheng, Guangqian Liu, Songtao Wang, Caihong Shen, Jin-Song Shi, Zheng-Hong Xu, Xiao-Juan Zhang

**Affiliations:** 1Key Laboratory of Industrial Biotechnology of Ministry of Education, School of Biotechnology, Jiangnan University, Wuxi 214122, China; 2National Engineering Research Center for Cereal Fermentation and Food Biomanufacturing, Jiangnan University, Wuxi 214122, China; 3School of Biotechnology, Jiangnan University, Wuxi 214122, China; 4National Engineering Research Center of Solid-State Brewing, Luzhou 646000, China; 5Jiangsu Engineering Research Center for Bioactive Products Processing Technology, Jiangnan University, Wuxi 214122, China

**Keywords:** sauce-aroma Baijiu, odor balance, compound abundance correlations, random forest classifier

## Abstract

Sauce-aroma Baijiu (SAB) is one of the most famous Baijius in China; SAB has more than 500 aroma compounds in it. However, the key aroma compound in SAB flavor remains unclear. Volatiles play an important role in SAB aroma and are highly correlated to SAB quality. In the present study, 63 volatile compounds were quantified among 66 SAB samples using gas chromatography with flame ionization detector (GC-FID). The authors analyzed odor contributions and volatile compound correlations in two quality groups of SAB samples. Moreover, an odor activity value (OAV) ratio-based random forest classifier was used to explain the volatile compound relationship differentiations between the two quality groups. Our results proved higher quality SABs had richer aromas and indicated a set of fruity-like ethyl valerate, green- and malt-like isobutyraldehyde and malt-like 3-methylbutyraldehyde and sweet-like furfural, had closer co-abundance correlations in higher quality SABs. These results indicated that the aroma and contributions of volatile compounds in SABs should be analyzed not only with compound odor activity values, but also the correlations between different aroma compounds.

## 1. Introduction

Chinese Baijiu is one of the six most famous distilled spirits in the world, and among all types of Baijiu, sauce-aroma Baijiu (SAB; also called Maotai-flavor Baijiu) is one of the most famous Baijiu in China [1]. SAB is made from sorghum and wheat using sophisticated fermentation techniques with multiple rounds of fermentation/distillation [2].

The complexity of SAB in flavors and compositions has been broadly realized, and a recent study showed that more than 500 volatile compounds can be detected in SAB. More than 100 important aroma compounds obtained via odor activity values (OAVs) were analyzed, and the ten aroma compounds which had the highest OAV values in SAB (ethyl butanoate, ethyl hexanoate, isopentyl hexanoate, ethyl isobutyrate, ethyl octanoate, ethyl isovalerate, ethyl propanoate, ethyl 2-methylbutanoate, butyl butyrate, propyl acetate) contributing in fruity, apple, sweet, grape, banana, pear, pineapple, floral and caramel odors [2]. However, the key aroma compound or a key set of aroma compounds representing SAB flavor or quality has not yet been revealed [3].

Different fermentation methods and varied environmental conditions result in the variation of flavors [4]. The discrimination between different types of Baijiu [5], the same type of Baijiu produced in different areas [6], and the discrimination of Baijiu with varied aging time [7] have been widely studied. For the aforesaid purposes, several classification methods were used such as partial least squares-discriminant analysis (PLS-DA) [8], random forest classification [9], and support vector machine (SVM) [10]. For example, Baijiu with different aging times possess varied organoleptic properties due to the change of volatile profile during storage [11], and ethyl oleate and long-chain fatty acid ethyl esters may be the markers for the aging time of Baijiu [12]. Cheng et al. [13] reported on an SVM quality discrimination model using mass spectrometry data which had high accuracy in strong-aroma Baijiu. However, quality grade discrimination of sauce-aroma Baijiu is limited [13].

Usually, volatile compounds with an OAV higher or equal to 1 are considered contributors to odor. Compounds having the highest OAVs were determined as the key odorants in Baijiu [14]. However, the organoleptic properties and flavor significances are not linear to the concentration of compounds [15], and it is worth noting that the volatile compounds interact with each other, resulting in the complexity of chemometric analysis, for example, the pickle-like odor had a positive correlation to the caramel-like odor [16], which suggested that compounds with different flavors might affect the overall flavor of Baijiu, and the sensory characteristics of SABs might be sensitive to the balance of these trace compounds.

Understanding of the aroma contribution of compounds in Baijiu is complicated due to the various perceptual interactions in the system. For example, high concentrations of 1-propanol and 2-phenylethanol were able to significantly inhibit the release of 3-methylbutyric acid in SABs [17]. Because of perceptual interactions, a mixture percept can be heterogeneous when several odors exist in the mixture [18]. Therefore, an additional methodology should be developed alongside the traditional aroma characterization method to describe the odor relationships in Baijiu. In this study, based on the chemometrics of volatiles of SAB samples with different grades, the authors focused on the balance of flavor compounds and the odor interactions in the SAB system (53% alcohol by volume). An algorism based on a random forest classifier for odor balances was developed, and multiple key features were identified via this algorism to illustrate key odor balances associated with the quality of SABs.

## 2. Materials and Methods

### 2.1. Baijiu Samples

A total of 66 SAB samples were collected from different distilleries in Maotai Town, Renhuai City, Guizhou Province, China. All 66 SAB samples were made via traditional soy sauce aroma-type solid-state fermentation with sorghum as raw material, in which 21 samples could be purchased in local markets and 45 samples were collected directly from factories. All samples had 53% alcohol by volume and were stored in a storage room under 15 °C for further analysis.

### 2.2. Chemicals

A total of 66 commercial standards (analytical reagent grade, ≥97% purity) were used. Ethyl acetate, acetic acid, Ethyl lactate, 1-propanol, acetal, acetaldehyde, 3-methyl butanol, furfural, ethyl palmitate, isobutanol, methanol, 3-methylbutyraldehyde, ethyl butyrate, ethyl propionate, meso-2,3-butanediol, butyrate, 1-butanol, (2R,3R)-(-)-2,3-butanediol, 2-butanol, hexanoic acid, ethyl hexanoate, 3-hydroxy-2-butanone, ethyl isovalerate, furfuryl alcohol, isobutyric acid, hexyl hexanoate, ethyl valerate, 1-hexanol, propionic acid, acetone, ethyl laurate, 2-methylbutyraldehyde, isoamyl acetate, 2-butanone, ethyl isobutyrate, ethyl phenylacetate, nonanoic acid, propyl acetate, isobutyraldehyde, ethyl octanoate, 1,1-diethoxy-2-methylbutane, ethyl linoleate, valeric acid, ethyl oleate, ethyl heptanoate, ethyl formate, phenylethyl alcohol, ethyl hexadecenoate, 1-pentanol, ethyl tetradecanoate, diethyloxymethane, 2-pentanone, 2-methyl butanol, isovaleric acid, ethyl nonanoate, 2-pentanol, 1,1-diethoxy-3-methylbutane, benzaldehyde, ethyl decanoate, heptanoic acid, octanoic acid, 2-ethoxy-5-methylfuran, 1-propanal, butyl hexanoate, ethyl phenylpropionate, and benzyl alcohol were purchased from Sigma-Aldrich (Shanghai, China).

### 2.3. Sensory Evaluation

Ten well-trained panelists including 4 national Baijiu panelists judged all SAB samples. Samples were then grouped into excellent grade, grade I, and grade II according to the China National Institute of Standardization GB/T 26760-2011. Briefly, SAB’s aroma, taste, and aftertaste were evaluated. For excellent grade SAB, the Baijiu body had an elegant fragrance with the sauce flavor taking the prominent position, the Baijiu body had plentiful integrity, the feeling in the mouth had mild-mannered coordination, and the feeling in the finish was prolonged. For grade I SAB, the Baijiu body had a pure and natural sauce flavor, the feeling in the mouth was well coordinated, and the finish was prolonged. For grade II SAB, the Baijiu body had a significant sauce flavor, the feeling in the mouth was mild, and the finish was relatively long.

### 2.4. Gas Chromatography with Flame Ionization Detector (GC-FID)

The analyses were performed on a GC 7890B gas chromatograph (Agilent Technology, Santa Clara, CA, USA) equipped with a flame ionization detector (FID). A DB-WAX column (60 m × 0.25 mm i.d., 0.25 μm film thickness, Agilent Technology, Santa Clara, CA, USA) was operated under programmed temperature conditions: 35 °C held for 8 min, 35–40 °C at 2.5 °C/min, 40–100 °C at 5 °C/min, 100–200 °C at 10 °C/min, 200–220 °C at 20 °C/min and held for 10 min (detector temperature of 250 °C, injector temperature of 230 °C), and nitrogen gas carrier gas (1.6 mL/min). Three internal standards were used in this study: amyl acetate (IS1, 10 μL, 10.33g/L), 2-methyl hexanoic acid (IS2, 10 μL, 14.00 g/L), and tertiary amyl alcohol (IS3, 10 μL, 8.05 g/L). The mixed standard solutions (1 mg/mL) were prepared in absolute ethanol and diluted with 53% (*v*/*v*) ethanol to the following concentrations: 50,000, 10,000, 5000, 1000, 500, 100, 50, 20, 10, and 1 mg/L. Each working standard solution (1 mL) was spiked with 10 μL IS1-3. Afterward, a 1 μL sample was then directly injected in a split mode (split ratio was 50:1). The LOD and LOQ values of standards were determined at their concentrations when their signal-to-noise ratios (S/N) were 3 and 10, and calibration curves were drawn using the response ratio between the target aroma and internal standards versus the ratio between their concentrations. Detailed information is provided in Appendix A. Every 1 mL SAB sample was spiked with 10 μL IS1-3, and then GC-FID quantification was performed the same way as the working standards. The signal was recorded and processed with OpenLab CDS (Agilent Technology, Santa Clara, CA, USA) software.

### 2.5. OAV Ratio-Based Random Forest Classifier

To evaluate the correlations between different volatile compounds, a ratio-based random forest classifier was used in this study programmed with Python packages including sklearn 0.23.2, pandas 1.1.4, and numpy 1.19.4. Three data tables are needed for the classifier including a table of compound quantities, a table of odor thresholds, and a table of odor descriptions. Briefly, (1) data were cleaned based on non-zero values and outliers; (2) the OAVs of metabolites were divided exhaustively and then logged by 2 to get the correlations for each set of compounds; (3) an optimized random forest classifier was used for group separations, and the parameters (number of trees, maximum depth, minimum of leaves) for the classifier were chosen with the GridSearchCV algorism together with 10-fold cross validation. The algorism used in this study can be found at GitHub with the following link: https://github.com/MiracleEaTu/ratio_based_corr_classifier, accessed on 5 October 2022.

### 2.6. Statistics

The significance between the two groups in this study was calculated using the Kruskal–Wallis test. A *p*-value less than 0.05 was considered significant. Principle component analysis (PCA) was analyzed with GraphPad Prism 9.0. A Pearson correlation coefficient greater than 0.6 of two odors was the criterion used for a connection (edge) between pairs of odors. The odor correlation network was visualized using Cytoscape software (v3.9.0, National Institute of General Medical Sciences, Bethesda, MD, USA) [19]. Other figures were visualized with the ‘ggplot2’ package in R 4.1.0.

Before the random forest classification, volatile compounds that had zero values in more than half of the samples were removed and outlier samples in each group were removed with local outlier factor algorism performed with the sklearn 0.23.2 package.

## 3. Results and Discussion

### 3.1. The Differences in Aroma Compounds between the Two Groups of SABs

According to the sensory evaluation, 39 samples were evaluated as excellent grade or grade I; in the present study, these samples were grouped as a relatively good (RG) group. Meanwhile, 27 samples were evaluated as grade II and these samples were grouped as a relatively poor (RP) group.

Based on the GC-FID result, 63 volatile compounds were quantified in at least 17 SAB samples, including 27 esters, 12 alcohols, 10 acids, 5 aldehydes, 4 alkanes, 3 ketones and 2 furans. It was reported that esters and alcohols contributed greatly to the aroma of SABs and there were 6 important skeleton aroma compounds (ethyl hexanoate, ethyl acetate, ethyl butyrate, 1-propanol, 2-phenylethanol, and 3-methylbutanol) in SAB [20]. In our study, all 6 important aroma compounds were quantified. However, only 3-methylbutanol had a statistical difference between the RG and RP groups and was higher in the RG group. This result indicated that although the SAB skeleton aroma compounds are important for SAB flavor, their concentration alone could not comprehensively present the quality of SABs.

To illustrate the aroma compound differentiations between the RG and RP groups, PCA was performed. The PCA result (Figure 1A,B) indicated a high similarity of aroma compound constructions between the RG and RP groups. Among 63 quantified compounds, 14 compound abundances were statistically different between the RG group and the RP group (Figure 1C). It was reported that two fruity-like odors (ethyl isobutyrate, ethyl isovalerate) gave additive or synergistic odor effects for Baijiu [21]. The result was consistent in our study; the concentrations of ethyl isobutyrate and ethyl isovalerate were both significantly higher in the RG group. Except for a fatty-like compound (ethyl linoleate), the other 13 compounds all had higher abundances in the RG group; these compounds mainly contributed in fruity and malt aromas. The higher abundance of aroma compounds detected in the RG group indicated that SAB in the RG group had a richer aroma and that could possibly be the reason for the sensory evaluation judgment. Interestingly, isobutyraldehyde, 3-methylbutanol, and 3-methylbutyraldehyde had a contribution to the pungency perception [22], although pungency evaluation was generally perceived as a challenging task, owing to its low reproducibility, subjectivity, and taste fatigue of assessors [23]. It is interesting to notice that RG grade is associated with a relatively higher pungency character.

### 3.2. The Correlation between the Grade of SABs and OAV of Volatiles

The OAVs of the identified compounds were calculated to evaluate the odor contributions between the two groups of SABs. Among 64 volatile compounds, 63 compounds reported odor descriptions in distilled liquor and 36 odor descriptors were summarized from these compounds. Forty-one compounds had an average OAV higher than 1 in the RG group (Table 1). These volatile compounds were regarded as contributors to the overall aroma profiles of SAB samples according to the classical Baijiu aroma characterization methods [24,25]. Here in this study, we used an accumulated OAV (OAV_sum_) for each odor descriptor as the profiles of odor compositions in SAB samples, and the overall OAV_sum_ contribution is shown in Appendix A. The PCA result of OAV_sum_ (Figure 2A,B) illustrated that the odor composition in the RG group and the RP group had a high similarity, which was consistent with the PCA result calculated with compound quantities. Among 36 odors, 8 odors were statistically different between the RG group and the RP group (Figure 2C). Not surprisingly, all OAV_sum_ of eight odors in the RG group were higher than those in the RP group. The richer aroma in the RG group not only contained bread-, sweet-, malt-, green-, and creamy-like odors, but also contained pungent-, rancid-, and fusty-like odors. Interestingly, the OAV_sum_ of the fruity-like odor did not show any significance between the two groups, whereas fruity-like aroma compounds were an important part in the SAB flavor profiles [26,27]. Aroma compounds in Baijiu had aroma compound interactions; thus, the authors believed odor interactions should be considered as an important factor to the perceptual quality, and therefore, the aroma compound balances should be taken into consideration for Baijiu grading and these interactions may explain the differentiations between the sensory evaluation and OAV_sum_ results.

### 3.3. Compound Co-Abundance Correlation Network of SAB of Two Grades

The authors applied a correlation network analysis based on Pearson’s correlation coefficient (Figure 3A), and 24 compounds were included in the network. The main cluster in the network centered on furfural, an important flavor compound that contributed in nutty-, sweet-, and bread-like odors in SABs [35]. Fruity-like volatile compounds take an important part in the correlation network, including 2-butanol, ethyl propanoate, ethyl isobutyrate, ethyl hexadecenoate, isoamyl acetate, and isobutanol, which were all correlated to furfural, and the average abundance of these compounds in the RG group were all higher than that in the RP group. It is worth noting that due to the esterification and hydrolyzation balance of ester/acid in SAB [36], a close association between several pairs of ester/acid in the correlation network was noticed, for example, the correlation between acetic acid and ethyl acetate, and the correlation between butyric acid and ethyl butyrate.

To better understand the differentiation of compound abundance correlation in the two groups, two correlation networks were made separately in the RG group and the RP group (Figure 3B,C). The core of the main cluster in the RG group was a combination of furfural, isobutyraldehyde, 3-methylbutyraldehyde, and isobutanol, whereas the core of the main cluster in the RP group was made up of isobutyraldehyde, 3-methylbutyraldehyde, 2-butanol, and propyl acetate. Isobutyraldehyde and 3-methylbutyraldehyde had strong correlation in both the RG group and the RP group. Furfural had more correlations with other compounds in the RG group than in the RP group (14 compared to 5). The authors also calculated the OAV_sum_ correlation network of two groups (Appendix A) and the result was interesting. The main OAV_sum_ cluster in the RG group was centered with a bread-like odor, however, the main OAV_sum_ cluster in the RP group was centered with creamy- and fatty-like odors. In addition, furfural contributed to the nutty-, sweet-, and bread-like odors. These results together indicated that furfural might be an important compound that may affect the SAB sensory, and the co-abundance of furfural with several compounds (e.g., 3-methylbutanol, isobutanol) could have a positive impact on SAB flavor. As a supplement to the perceptual interactions, the co-abundance network analysis here provides a new approach to unravel the complexity of the Baijiu system.

### 3.4. Important Odor and Compound Sets Selected by OAV Ratio-Based Random Forest Classification between the Two Quality Groups of SABs

Based on the compound composition results, the authors hypothesized that odor balances play an important role in SAB quality, and the ratio of compounds with different descriptors might be able to reflect the odor balance and, therefore, could be applied as candidate input parameters for classifiers of SAB quality grading. It is reported that the aroma other than the saltness of Douchiba, a Chinese traditional soy-fermented condiment, depends not only on key compounds but also on a “critical balance” or a “weighted concentration ratio” of 60 compounds [37]. Here, we adapted the concentration ratios and calculated the balances of aroma compounds in SAB. A random forest classifier is suitable for non-linear classifications [38] and was used for grouping the samples.

Specifically, OAV ratios of aroma compounds in two sets of odor descriptions were generated and logged by 2, as well as the OAV_sum_ of odor descriptions. A total of 1097 logged OAV ratios were generated, and the dataset was regarded as the classifier input.

Then, the OAV ratio-based random forest classifier was performed, and the results are shown in Figure 4; the classifier gave us a reasonable 83.33% accuracy. The top 5 features ordered by importance were OAV_sum_ of green-like odor/OAV_sum_ of sweet-like odor, OAV_sum_ of green-like odor/OAV_sum_ of strawberry-like odor, OAV of ethyl valerate/OAV of isobutyraldehyde, and OAV of isobutyraldehyde/OAV of ethyl valerate. These ratios logged by 2 were all statistically different between the RG group and the RP group.

The authors also used an OAV dataset (Appendix A) and a combined dataset including an OAV table after z-score normalization and the 1097 logged ratios (Appendix A) for the classification. In the OAV dataset classification, the top 5 features ordered by importance in the classifier were propyl acetate, isobutanol, furfuryl alcohol, 3-methylbutanol, and 2-butanol, and the top 10 features in the classifier contributed to 6 odors, including fruity-, sweet-, pungent-, rancid-, sweaty-, and malt-like odors. The result of the combined dataset classification was basically a combination of the OAV classification and the OAV ratio-based random forest classification.

The ratio between ethyl valerate and isobutyraldehyde had high feature importance in the classifier. Ethyl valerate did not have a statistical difference between the RG group and the RP group, it had fruity-, apple- and strawberry-like odors [16], and these odors did not show differentiation between the RG group and the RP group either (Figure 2C). Isobutyraldehyde, which was significantly higher in abundance in the RG group, had pungent-, malt-, and green-like odors [29], and these odors were all included in the significantly increased odors in the RG group. Notably, the OAV_sum_ of the green-like odor divided by the OAV_sum_ of the strawberry-like odor was also an important feature in the classifier. Additionally, as shown in Figure 3, in the RG group, the green-like odor was positively correlated to the strawberry-like odor, but in the RP group, there was no correlation between the green-like odor and the strawberry-like odor. This result indicated a potential correlation between the green-like odor and the strawberry-like odor, which was mainly contributed by isobutyraldehyde and ethyl valerate.

These results together with the OAV-based random forest classifier results and the odor correlations gave us a new insight into odor balances in the two groups of SABs. The classifier is suitable for compounds with only relative abundances such as peak areas detected from mass spectrometry. There are still several limitations to our OAV ratio-based classifier. For example, some aroma compounds have multiple odor descriptions, such as ethyl valerate, isobutyraldehyde, and ethyl hexanoate. The OAVs of these compounds were used multiple times in our odor correlation and group classifier calculations, which may duplicate the important features in the results.

### 3.5. Correlations of Key Compound Sets in SABs

Based on our OAV ratio-based classification result and content OAV classification, we selected 25 possible compounds that either had differentiation in abundance or showed high feature importance in at least one classification (Appendix A).

The differentiations of sets of volatile compounds between the RG and RP groups were identified (Figure 5). The interesting compound set of ethyl valerate and isobutyraldehyde had a positive correlation in both RG and RP groups. Isobutyraldehyde also had a positive correlation with 3-methylbutyraldehyde and furfural in both the RG and RP group. It is worth noting that in the RG group, the correlation coefficient scores between isobutyraldehyde and 3-methylbutyraldehyde and the correlation coefficient scores between isobutyraldehyde and furfural were above 0.9, representing the balances between these three compounds in SABs. Furfural had nutty-, sweet- and bread-like odors, whereas isobutyraldehyde had pungent-, malt- and green-like odors, and 3-methylbutyraldehyde had grass- and green-like odors. It was reported that 3-methylbutyraldehyde was one of the key aroma compounds in SAB [2], as well as the furfural content in SABs, and was more than those in strong-flavor Baijiu and light-flavor Baijiu [35]. The co-abundance relationship of these four volatile compounds had a p-value less than 0.05 but different correlation coefficient values in all SAB samples, which gave us a new insight that the combination of ethyl valerate, 3-methylbutyraldehyde, isobutyraldehyde, and furfural may be an important aroma set for SABs. In addition, the relationship between furfural and the other two compounds also supports that the off-odor compounds also play important roles in SAB flavors.

## 4. Conclusions

In this study, the authors quantified 64 aroma compounds in 66 SAB samples using a GC-FID platform; samples were grouped into two quality groups based on their sensory evaluation result. The authors used aroma compound correlations and OAV ratio-based classification to illustrate the odor balance differentiations between the RG and the RP groups. The results showed the OAV-based random forest classification and the OAV ratio-based random forest classification had the same accuracy but different precision. According to the OAV ratio-based random forest classification and compound correlations in the two quality groups of SABs, the balance of fruity-like ethyl valerate, green- and malt-like isobutyraldehyde, and malt-like 3-methylbutyraldehyde and sweet-like furfural might be one of the most important aroma compound balances sets in SAB. Though this study illustrated a new insight into aroma compound correlation, future studies are needed to illustrate the mechanism of aroma compound interactions and assess the impact of compound balance via sensory tests. The classifier established here using the compound OAV ratio as an additional dataset can be improved to be more suitable in future flavor studies.

## Figures and Tables

**Figure 1 foods-11-03916-f001:**
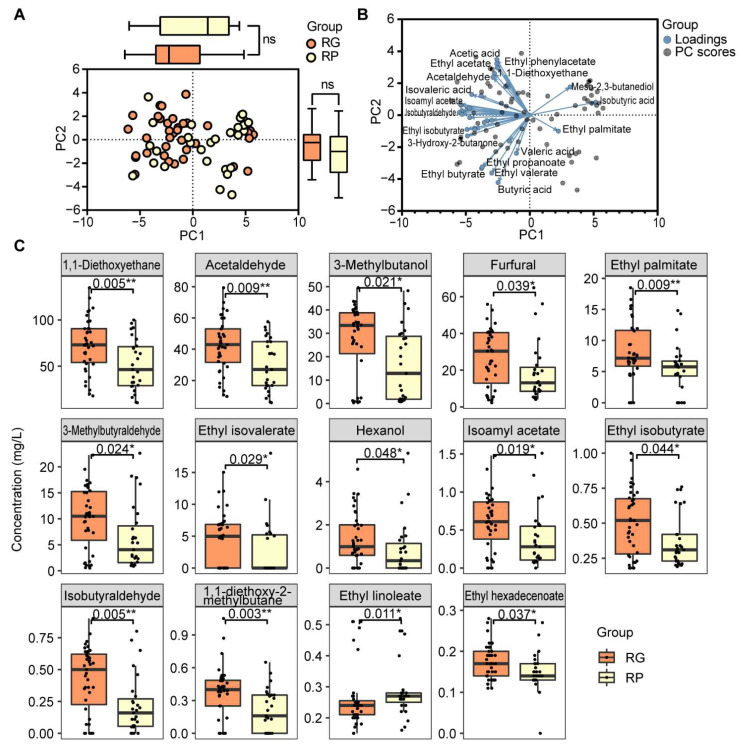
The PCA result and the differential abundances of aroma compounds in the RG group and the RP group. (**A**) PCA result representing the differentiation of the aroma compound abundances between the RG group and RP groups. The orange dots represent samples from the RG group, and the yellow dots represent samples from the RP group. The boxplot at the top represents the PC1 values in two groups, and the boxplot on the right represents the PC2 values in two groups. The orange boxes represent the RG group, and the yellow boxes represent the RP group. (**B**) PCA biplot of the aroma compound abundances between the RG and RP groups. The blue dots represent the compound loadings, and the grey dots represent the samples. (**C**) Boxplots of compounds having statistically different abundances between the RG and RP groups. The orange boxes represent the compound abundances in the RG group, and the yellow boxes represent the compound abundances in the RP group. The significances were calculated via the Kruskal–Wallis test and a p-value less than 0.05 was considered significant. *: *p*-value ≤ 0.05, **: *p*-value ≤ 0.01, ns: not significant.

**Figure 2 foods-11-03916-f002:**
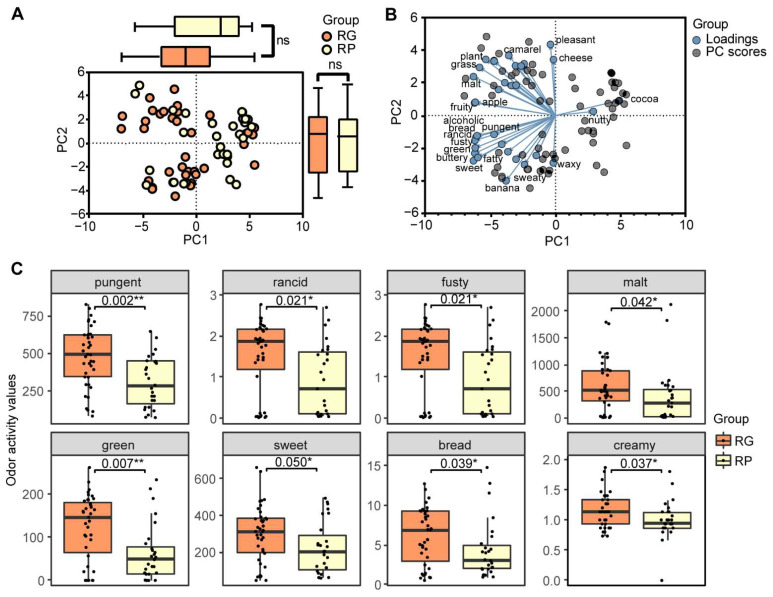
The PCA result and the differences of accumulated OAV in the RG group and the RP group. (**A**) PCA result representing the differentiation of the OAV_sum_ between the RG group and RP groups. The orange dots represent samples from the RG group, and the yellow dots represent samples from the RP group. The boxplot on the top represents the PC1 values in two groups, and the boxplot on the right represents the PC2 values in two groups. The orange boxes represent the RG group, and the yellow boxes represent the RP group. (**B**) PCA biplot of the OAV_sum_ between the RG and RP groups. The blue dots represent the odor loadings, and the grey dots represent the samples. (**C**) Boxplots of odors having statistically different OAV_sum_ between the RG and RP groups. The orange boxes represent the OAV_sum_ in the RG group, and the yellow boxes represent the OAV_sum_ in the RP group. The significances were calculated via the Kruskal–Wallis test and a *p*-value less than 0.05 was considered significant. *: *p*-value ≤ 0.05, **: *p*-value ≤ 0.01, ns: not significant.

**Figure 3 foods-11-03916-f003:**
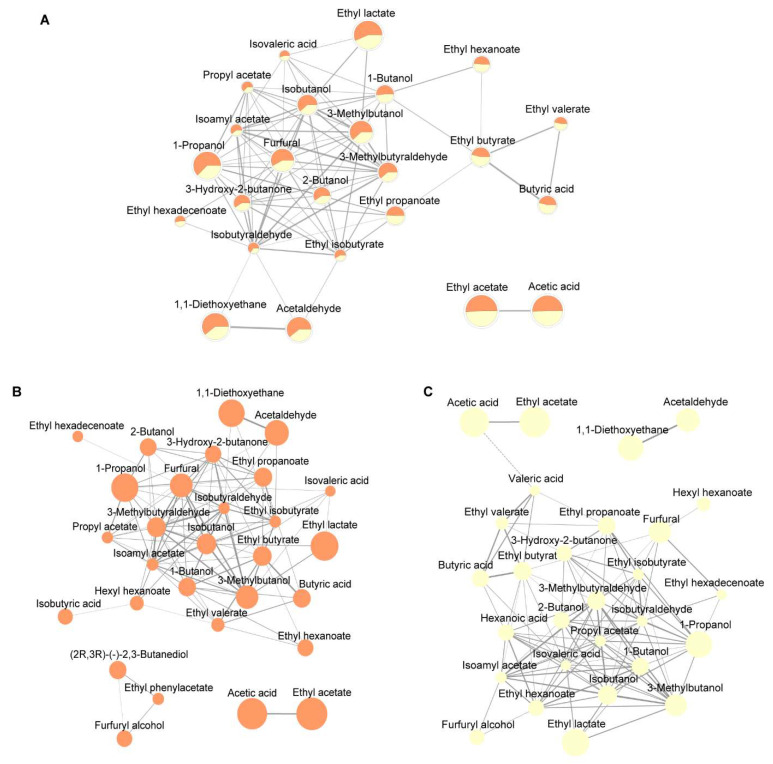
Compound co-abundance correlation network calculated with correlation coefficient of two groups. (**A**) Compound co-abundance correlation network in the total dataset. The pie charts showing the nodes represent the average compound abundance in two groups. The orange pies represent the compound average abundance in the RG group, and the yellow pies represent the compound average abundance in the RP group. (**B**) Compound co-abundance correlation network in the RG group. (**C**) Compound co-abundance correlation network in the RP group. The solid edges represent positive correlations between the compounds, and the dashed edge represents a negative correction between the compounds. The size of the nodes represents the logged average compound abundances in SAB, and the width of the edges represents the correlation coefficient value of two compound abundances; wider edges represent stronger correlations.

**Figure 4 foods-11-03916-f004:**
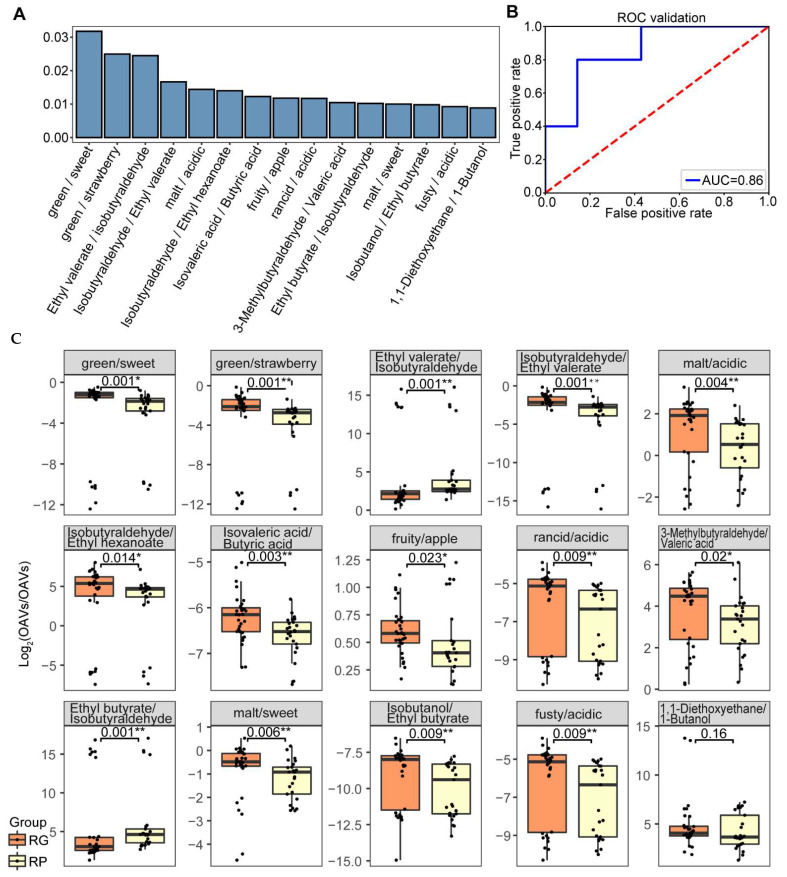
Random forest classification using logged OAV ratio dataset. (**A**) Top 15 features ordered by importance in the OAV ratio-based random forest classifier. (**B**) ROC curve of the OAV ratio-based random forest classifier. (**C**) Boxplots of OAV ratios logged by 2 represent the 15 features in the OAV ratio-based random forest classifier between the RG group and the RP group. The orange boxes represent the OAV ratios logged by 2 in the RG group, and the yellow boxes represent the OAV ratios logged by 2 in the RP group. The significances were calculated via the Kruskal–Wallis test, and a p-value less than 0.05 was considered significant. *: *p*-value ≤ 0.05, **: *p*-value ≤ 0.01.

**Figure 5 foods-11-03916-f005:**
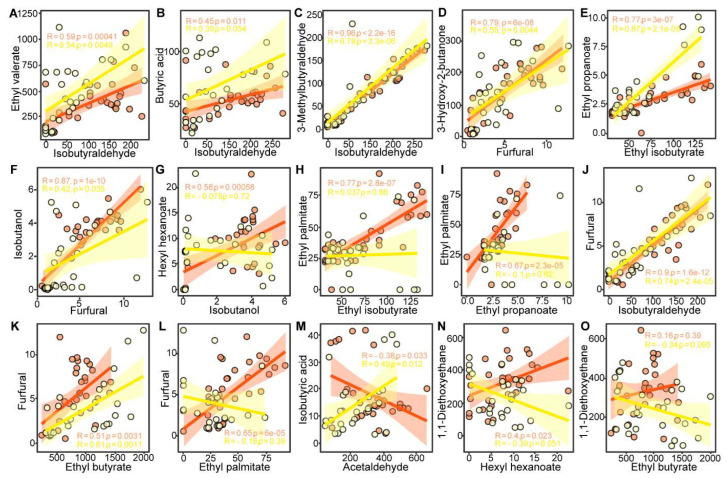
Correlations of key compound OAVs in the RG and the RP groups. (**A**) The correlation between isobutyraldehyde and ethyl valerate. (**B**) The correlation between isobutyraldehyde and butyric acid. (**C**) The correlation between isobutyraldehyde and 3-methylbutyraldehyde. (**D**) The correlation between furfural and 3-hydroxy-2-butanone. (**E**) The correlation between ethyl isobutyrate and ethyl propanoate. (**F**) The correlation between furfural and isobutanol. (**G**) The correlation between isobutanol and hexyl hexanoate. (**H**) The correlation between ethyl isobutyrate and ethyl palmitate. (**I**) The correlation between ethyl propanoate and ethyl palmitate. (**J**) The correlation between isobutyraldehyde and furfural. (**K**) The correlation between ethyl butyrate and furfural. (**L**) The correlation between ethyl palmitate and furfural. (**M**) The correlation between acetaldehyde and isobutyric acid. (**N**) The correlation between hexyl hexanoate and 1,1-diethoxyethane. (**O**) The correlation between ethyl butyrate and 1,1-diethoxyethane. The orange dots represent the OAV values of a sample in the RG group, and the yellow dots represent the OAV values of a sample in the RP group. The orange lines with light orange backgrounds represent the trend lines and standard deviations of OAV values in the RG group and the yellow lines with canary backgrounds represent the trend lines and standard deviations of OAV values in the RP group The R values represent correlation coefficient values, and the *p* values represent the significance of the correlations.

**Table 1 foods-11-03916-t001:** Aroma compounds detected in SABs with OAV higher than one.

Compound Name	Odor Threshold (μg/L)	Odor Descriptor	Reference	OAV			
				(RG Group)		(RP Group)	
				Average	Std.	Average	Std.
Ethyl isovalerate	6.89	Fruity	[3]	6484.69	6010.65	3683.28	6341.29
Ethyl butyrate	82.00	Apple, fruity	[28]	886.24	362.32	986.18	509.55
Ethyl valerate	27.00	Fruity, apple, strawberry	[28]	414.43	228.42	484.91	317.67
2-Methylbutyraldehyde	17.00	Grass, plant, malt	[28]	372.55	387.49	267.10	437.57
Acetaldehyde	1200.00	Pungent	[3]	351.96	141.28	247.22	136.93
1,1-Diethoxyethane	2090.00	Fruity	[20]	347.55	135.50	246.23	132.20
Ethyl octanoate	12.90	Pineapple, fruity, floral	[3]	250.84	315.34	233.13	329.38
3-Hydroxy-2-butanone	259.00	Fatty, buttery, sweet	[3]	174.26	107.57	123.27	97.07
Isobutyraldehyde	34.69	Pungent, malt, green	[29]	121.43	75.02	62.88	65.93
3-Methylbutyraldehyde	980.00	Grass, malt	[28]	102.88	61.73	65.08	63.47
Ethyl isobutyrate	57.50	Fruity, sweet	[3]	87.45	40.15	65.31	32.61
Ethyl acetate	32,600.00	Fruity, buttery, orange	[3]	67.09	12.75	64.51	15.41
Isoamyl acetate	93.93	Fruity	[3]	63.74	39.15	41.20	40.46
Butyric acid	964.00	Sweaty, acidic, mud	[28]	57.50	18.50	66.69	30.54
2-Methyl butanol	16.00	Cocoa, nutty	[3]	45.35	129.84	60.65	130.94
Ethyl palmitate	2000.00	Nutty	[28]	40.57	23.27	100.01	376.23
2-pentanone	194,000.00	Wine	[20]	26.56	42.05	35.71	89.37
1-Butanol	2730.00	Alcoholic, sweet	[3]	22.01	14.26	19.85	19.79
Hexanoic acid	13,300.00	Sweaty	[28]	19.80	16.65	14.14	10.97
Isobutyric acid	2300.00	Sweaty	[30]	17.60	9.91	14.56	10.63
1-Propanol	54,000.00	Alcoholic	[3]	17.53	14.53	13.35	16.60
Ethyl laurate	500.00	Fruity, floral	[21]	15.16	17.58	9.13	13.77
Ethyl phenylacetate	407.00	Rosy, floral, honey	[3]	11.94	8.01	9.09	8.12
Acetic acid	160,000.00	Vinegar, acidic	[3]	10.10	2.10	9.82	2.03
Hexyl hexanoate	1890.00	Fruity	[28]	8.79	5.61	7.63	4.93
Ethyl lactate	128,000.00	Fruity	[3]	8.34	3.56	6.81	4.21
Valeric acid	389.00	Sweaty	[31]	7.15	9.72	5.49	3.24
Furfural	44,000.00	Nutty, sweet, bread	[3]	6.10	3.56	4.28	3.68
(2R,3R)-(-)-2,3-Butanediol	12,000.00	Celery, fruity	[32]	4.26	2.10	4.32	2.21
Ethyl propanoate	19,019.00	Banana, fruity	[3]	3.50	1.86	3.61	2.46
Isobutanol	40,000.00	Fruity	[28]	3.15	1.84	2.01	2.06
Ethyl hexanoate	13,200.00	Fruity, floral, sweet	[28]	2.72	2.33	2.51	2.09
Furfuryl alcohol	12,323.00	Caramel, sweet	[31]	2.51	1.59	2.43	2.32
Hexanol	5370.00	Floral	[3]	2.48	2.21	1.60	2.38
Ethyl myristate	500.00	Sweet, waxy, violet orris	[3]	2.44	3.46	2.14	2.56
Propyl acetate	2700.00	Fruity, sweet	[33]	1.61	1.19	1.05	1.07
3-Methylbutanol	179,000.00	Rancid, fusty, malt, green	[3]	1.51	0.87	0.92	0.87
1,1-Diethoxy-3-methylbutane	323.00	Fruity, fatty	[34]	1.44	2.72	0.86	2.87
Ethyl hexadecenoate	1500.00	Fruity, creamy	[33]	1.16	0.28	1.00	0.31
2-Butanol	50,000.00	Fruity, alcoholic	[28]	1.06	0.88	0.77	0.67
Nonanoic acid	3559.00	Cheese	[3]	1.00	1.69	1.40	1.78

## Data Availability

The algorism used in this study can be found at GitHub with the following link: https://github.com/MiracleEaTu/ratio_based_corr_classifier, accessed on 5 October 2022.

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
