# Peer review of "Volatile Compound Abundance Correlations Provide a New Insight into Odor Balances in Sauce-Aroma Baijiu"

_foods, 2022, doi:10.3390/foods11233916_

Round 1

Reviewer 1 Report

The manuscript titled: “Volatile compound abundance correlations provide a new insight into odor balances in sauce-aroma Baijiu” describes an analysis of 66 commercial Baijiu’s to quantify 66 aroma compounds using GC-FID. The purpose was to gain a better understanding of the aroma contribution of these compounds in Baijiu.

This manuscript is well-written, and the authors did a good job in presenting their findings. I did have some minor comments to help provide additional clarification and supporting information.

Comments:

1.     There are some minor grammatical/syntax errors throughout the manuscript. I would recommend a thorough read through to address them.

2.     66 standards are mentioned, but 64 compounds are mentioned in the concluding remarks? Were 2 of the compounds not identified in any of the samples?

3.     Would suggest listing all 66 commercial standards in table format, with retention times and LOQ and LOD.

4.     Was a standard curve made to determine LOQ and LOD? What was the concentration range of the volatile compounds identified?  If not, how was the data normalized to make comparisons among samples?

Author Response

#1. There are some minor grammatical/syntax errors throughout the manuscript. I would recommend a thorough read through to address them.

Answer: Thank you for your comment. We read the manuscript and revised the grammatical/syntax errors.

#2. 66 standards are mentioned, but 64 compounds are mentioned in the concluding remarks? Were 2 of the compounds not identified in any of the samples?

Answer: Thank you for your comments. There were two compounds (benzyl alcohol, octanoic acid) only quantified in less than 6 samples, and the quantity was very close to the LOQ, so we decided to remove them.

#3. Would suggest listing all 66 commercial standards in table format, with retention times and LOQ and LOD.

Answer: Thank you for your comment. We apologize for missing this information in the method part. We modified method part and added this information in the supplementary table.

#4. Was a standard curve made to determine LOQ and LOD? What was the concentration range of the volatile compounds identified?  If not, how was the data normalized to make comparisons among samples?

Answer: Thank you for your comment. We apologize for missing this information in the method part. We made mixed standard curves before ran actual samples. The range of compound quantities suited our sample. We modified method part and added this information in the supplementary table.

Reviewer 2 Report

Although scientifically the present work seems interesting I believe that the analytical part suffers a great deal.  According to the authors, 64 aroma compounds were quantified in 66 SAB samples using GC-FID.

Firstly, before every quantification procedure, a qualification process precedes, which in the present work is absent. When samples rich in ingredients have to be analyzed the most appropriate procedure is to apply GC-MS instrumentation along with authentic samples injection or even better co-injection.

Secondly, the quantification procedure is not presented or described in the text, as it should. Did authors build curves?  

Thirdly, since SAB is an aqueous alcoholic drink how did the authors perform the above-mentioned analysis? Did they inject directly volumes of the SAB samples into the injection port?

Author Response

#1 Firstly, before every quantification procedure, a qualification process precedes, which in the present work is absent. When samples rich in ingredients have to be analyzed the most appropriate procedure is to apply GC-MS instrumentation along with authentic samples injection or even better co-injection.

Answer: Thank you for your comments. Co-injection of GC-MS and GC-FID could quantify compounds properly, GC-MS/O system is even better. During our study design period, we believed that GC-FID quantitative method is enough for a volatile compound correlation study if the compound standard curve meets its requirement. The deviation of compound quantities will affect the OAV values and ratios between compounds, but will have less effect on compound correlations and ratios correlations.

#2 Secondly, the quantification procedure is not presented or described in the text, as it should. Did authors build curves? 

Answer: Thank you for pointing this out. We apologize for missing this information in the method part. We did build standard curves and now it is added, the related information is added as supplementary table.

#3 Thirdly, since SAB is an aqueous alcoholic drink how did the authors perform the above-mentioned analysis? Did they inject directly volumes of the SAB samples into the injection port?

Answer: Thank you for pointing this out. We inject 1 μL SAB sample directly into inlet in this study, and it is added in the method part.